# Feature-Attending Recurrent Modules for Generalization in Reinforcement Learning

**Wilka Carvalho**[*]                                                          *wcarvalho@g.harvard.edu*
*Kempner Institute for the Study of Natural and Artificial Intelligence*
*Harvard University*

**Andrew K. Lampinen**                                                          *lampinen@google.com*
**Kyriacos Nikiforou**                                                          *knikiforou@google.com*
**Felix Hill**                                                                  *felixhill@google.com*
**Murray Shanahan**                                                            *mshanahan@google.com*
*Google DeepMind*

**Reviewed on OpenReview:** *https://openreview.net/forum?id=j4y3gN7VtW*

## Abstract

Many important tasks are defined in terms of objects. To generalize across these tasks, a reinforcement learning (RL) agent needs to exploit the structure that the objects induce. Prior work has either hard-coded object-centric features, used complex object-centric generative models, or updated state using local spatial features. However, these approaches have had limited success in enabling general RL agents. Motivated by this, we introduce "Feature-Attending Recurrent Modules" (FARM), an architecture for learning state representations that relies on simple, broadly applicable inductive biases for capturing spatial and temporal regularities. FARM learns a state representation that is distributed across multiple modules that each attend to spatiotemporal features with an expressive feature attention mechanism. We show that this improves an RL agent's ability to generalize across object-centric tasks. We study task suites in both 2D and 3D environments and find that FARM better generalizes compared to competing architectures that leverage attention or multiple modules.

## 1 Introduction

Objects are key to real-world tasks. For example, a self-driving car needs to represent the movements of other cars, and a household robot needs to recognize and use kitchen items. In order to generalize across tasks with objects, a reinforcement learning (RL) agent should capture and exploit object-induced structure present across the tasks.

One way to capture this structure is in an agent's state representation. Unfortunately, flexibly capturing objects in a state representation is challenging because an objects have many dimensions that can vary. Consider a household robot tasked with cooking. Completing the task might require memory about objects that range in size, shape, and color (e.g. a stove vs. a tomato). Additionally, objects in motion might require that the agent represent temporal information about the objects. It is unclear how to best incorporate objects into a state representations to enable generalization.

Prior work has attempted to capture object-induced task structure by hand-designing object-centric state features (Diuk et al., 2008; Carvalho et al., 2021; Borsa et al., 2018; Marom & Rosman, 2018). The "COBRA" agent (Watters et al., 2019) avoids hand-designing features by learning an object-centric generative model. However, these methods are limited in their generality because they rely on relatively strong inductive biases. For example, COBRA relies on environments being fully-observable and objects having regular shapes to

---

[*]Work done during internship. Codebase: https://github.com/wcarvalho/farm.

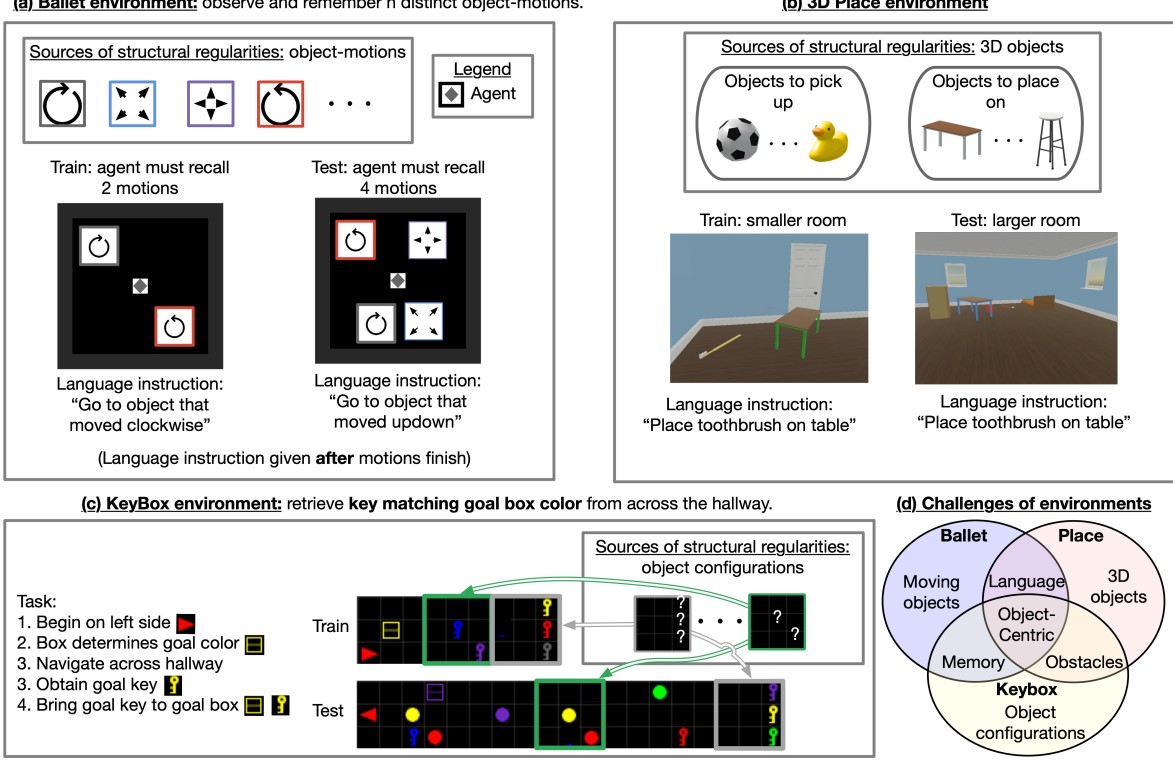

Figure 1: **Three environments with different structural regularities induced by objects**. In the Ballet environment, tasks share regularities induced by object motions; in the KeyBox environment, they share regularities induces by object configurations; and in the Place environment, tasks share regularities induces by 3D objects. The Ballet and KeyBox environments pose learning challenges for long-horizon memory and require generalizing to more objects. The KeyBox and Place environments pose learning challenges in obstacle navigation and requires generalizing to a larger map. Videos of our agent performing these tasks: https://bit.ly/3kCkAqd. See §3 for a description of the problem setting.

learn representations by predicting object segmentations. We focus on weak inductive biases in order to maximize an architecture's flexibility.

Objects can be described by subsets of features over space and time. We conjecture that weak inductive biases for capturing subsets of features over space and time may enable agents that can flexibly incorporate objects into state across a wide range of environments.

We propose *Feature Attending Recurrent Modules (FARM)*, a simple but flexible architecture for learning state representations when tasks share object-induced structural regularities. FARM learns state representations that are *distributed* across multiple, smaller recurrent modules. To help motivate this, consider word embeddings. A word embedding can represent more information than a one-hot encoding of the same dimension because subsets of dimensions can coordinate activity to represent different patterns of word usage. Analogously, learning multiple modules enables FARM to coordinate subsets of modules to represent different temporal segments in an agent's experiences. To capture general object-induced patterns, modules select observation information to update with by applying a mask to the channels of spatiotemporal observation features.

We study FARM across three diverse object-centric environments, each with their own suite of tasks that share object-induced structural regularities. Tasks in the Ballet environment share regularities induced by object motions; tasks in the Place environment share regularities induced by navigating towards and around 3D objects; and tasks in the KeyBox environment share regularities induced by object configurations. These environments present a number of challenges. First, their state-space grows exponentially with the number of

objects. The more distractor objects an environment has, the larger the chance an object will obstruct an agent's path. This requires learning a policy that can navigate around distractor-based obstacles. When task objects appear in sequence, this can require long-horizon memory of object information (e.g. of goal information). Finally, tasks defined by language can require an agent learn a complex mapping (e.g. to object motions and to irregular shapes in our tasks). Across these environments, we study an agent's ability to recombine object-oriented memory, obstacle-avoidance, and navigation to longer tasks with more objects.

We compare against methods with weak inductive biases for enabling objects to emerge in a state representation. Recent work has shown that spatial attention is a simple inductive bias for strong performance on object-centric vision tasks because it enables attending to individual objects (Greff et al., 2020; Locatello et al., 2020; Goyal et al., 2020a). Thus, we compare against recent RL agents that leverage spatial attention for object-centric state-update functions (Goyal et al., 2020b; Mott et al., 2019).

**Our core contribution is** to show that we can improves an RL agent's ability to generalize to out-of-distribution tasks by having multiple modules attend to spatiotemporal features with feature attention. We expand on this below:

1. FARM leverages multiple modules that each apply feature-wise attention to spatiotemporal features. This enables generalizing (a) memory to longer combinations of object motions (§5.1); (b) navigation to 3D objects in larger environments (§5.2); and (c) memory of goal information to longer tasks with more distractors (§5.3).
2. Competing methods have modules which leverage spatial attention, which has been shown to enable object-centric state updates. Across diverse object-centric RL tasks, we find that spatial attention has mixed benefits and can interfere with the benefits of learning multiple modules.
3. We hypothesize that FARM enables an RL agent to generalize to combinations of its experience by representing different temporal segments across subsets of modules (see Figure 2). In §5.3.1, we analyze FARM and provide evidence that object-induced temporal regularities are indeed represented across subsets of modules.

## 2 Related work on generalization in deep RL

The key question for generalization is how to capture structure in the problem in a flexible way. How much structure do you build in? How much do you let the agent discover? Some work takes a data-driven approach (Tobin et al., 2017; Packer et al., 2018; Hill et al., 2020; Justesen et al., 2019). Others have a policy that captures task structure with either hierarchical RL (Oh et al., 2017; Zhang et al., 2018; Sohn et al., 2018; 2021; Brooks et al., 2021) or successor features (Borsa et al., 2018; Barreto et al., 2020). A final strand focuses on learning invariant representations (Higgins et al., 2017; Chaplot et al., 2018; Lee et al., 2020; Zhang et al., 2021) or building in inductive biases (Mott et al., 2019; Goyal et al., 2020b). In this work we focus on weak inductive biases for capturing structure. Below we review approaches most closely related to ours.

**Generalizing across object-centric tasks** dates back at least to object-oriented MDPs (Džeroski et al., 2001; Diuk et al., 2008) which enabled generalization by representing dynamics with logical object attributes (Kansky et al., 2017; Marom & Rosman, 2018). Successor features have also leveraged objects for generalization by formulating rewards as linear with object-centric features (Borsa et al., 2018; Barreto et al., 2020). A common thread among these directions is that they relied on hand-designed object features. Watters et al. (2019) avoided hand-designing features by learning an object-centric generative model (Burgess et al., 2019). However, they focused on fully-observable top-down environments with regular shapes, which allowed them to predict future object masks. This is incompatible with our environments. While research on object-centric models (Kabra et al., 2021; Zoran et al., 2021) has progressed, these methods commonly add training complexity (more objective terms, extra modules, etc.) and make stronger assumptions (e.g. on the number of objects). We differ from this work because we focus on simple, broadly applicable inductive biases for capturing object-induced task regularities.

**Generalizing with feature attention** has also been studied by Chaplot et al. (2018). They showed that mapping language instructions to masks over the channels of observation features enabled generalization to language instructions with new feature combinations. While FARM also learns a mask over observation

features, our work has two important differences. First, we develop a multi-head version where different recurrent modules produce their own masks. This enables FARM to leverage this form of attention in settings where language instructions don't indicate what to attend to (this is true in 2/3 of our tasks). Second, we are the first to show that feature attention enables generalizing memory of object motions and of goal information to longer tasks (Figure 4 and Figure 6, respectively).

**Generalizing with top-down spatial attention**. Most similar to FARM are the Attention Augmented Agent (AAA) (Mott et al., 2019) and Recurrent Independent Mechanisms (RIMs) (Goyal et al., 2020b). Both are recurrent architectures that leverage spatial attention to learn an object-centric state-update function. Both showed generalization to novel distractors. The major difference between AAA, RIMs, and FARM is that FARM attends to an observation with feature attention as opposed to spatial attention. Our experiments indicate that spatial attention has limited utility in updating state during reinforcement learning of tasks defined by object motions (Figure 4) or 3D objects (Figure 5). In terms of modularity, we also show different results from RIMs who showed that their modules "specialize". Our experiments suggest that in FARM, a modular state instead leads subset of modules to *jointly* represent regularities in an agent's experience (§5.3.1).

## 3 Problem setting

We study generalization across tasks within deterministic, partially-observable, pixel-based environments. Within an environment, a task is defined by a Partially Observable Markov decision processes (POMDP): $\mathcal{M} = \langle \mathcal{S}, \mathcal{A}, \mathcal{O}, R, T, \psi \rangle$. $\mathcal{S}$ corresponds to environment states, $\mathcal{A}$ corresponds to actions that agent can take, $\mathcal{O}$ corresponds to the agent's observations, $r = R(s, a) \in \mathbb{R}$ is the reward function, $s' = T(s, a) \in \mathcal{S}$ is the environment transition function, and $o = \psi(s) \in \mathcal{O}$ is an observation function that maps the underlying environment state to an RGB image.

We seek an RL agent that learns to perform tasks by finding a policy $\pi$ that maximizes the expected discounted sum of rewards it obtains starting at a state $s$: $V(s) = \mathbb{E}\left[\sum_{t=0}^{\infty} \gamma^t R(S_t, A_t)\right]$—also known as the *value* of a state. In a POMDP, the agent doesn't have access to the environment state. A common strategy is to instead learn an *"agent state"* representation, $s_t^A$, that compresses the full history $(o_1, a_1, r_1, \ldots, a_{t-1}, o_t)$ into a sufficient statistic suitable for selecting actions. The agent state is commonly learned with a recursive function $s_t^A = \eta(o_t, a_{t-1}, r_{t-1}, s_{t-1}^A)$.

**Object-induced structural regularities**. We study object-centric environments, where objects induce structural regularities across tasks in the reward functions $R$, transition functions $T$, and observation functions $\psi$. For example, consider the KeyBox environment in Figure 1 (c). First, $R(s, a)$ always specifies the goal key based on a goal box. Second, whenever the agent has to navigate around an obstacle (see Figure 2, b), the agent always sees the sprite it controls move closer to an object and then around it. This is true regardless of *where* in the hallway the agent observes the obstacle because of regularities in the transition function $T(s, a)$ and observation function $\psi(s)$. We want an agent that captures these regularities in its representation for state to enable zero-shot generalization to new tasks.

## 4 Architecture: FARM

We propose a new architecture, "Feature Attending Recurrent Modules" (FARM) for learning an agent's state representations when an environment has object-induced structural regularities. We provide an overview of the architecture in Figure 2. Instead of representing agent state with a single recurrent function, FARM learns a state representation that is distributed across $n$ recurrent functions $\{\eta^k\}_{k=1}^n$, which we call modules (Figure 2, a). Distributing state across modules allows subsets of modules to jointly represent different regularities in the agent's experience (Figure 2, b). We hypothesize that having subsets of modules represent different regularities in the agent's experience enables the agent to flexibly recombine its experience for generalization.

At each time-step $t$, each module updates with both observation features and information from other modules. First, the agent computes observation features with a recurrent observation encoder, $\boldsymbol{Z}_t = \phi(o_t, \boldsymbol{Z}_{t-1})$. Afterward, each module creates a *query* vector by combining its previous module-state with the previous

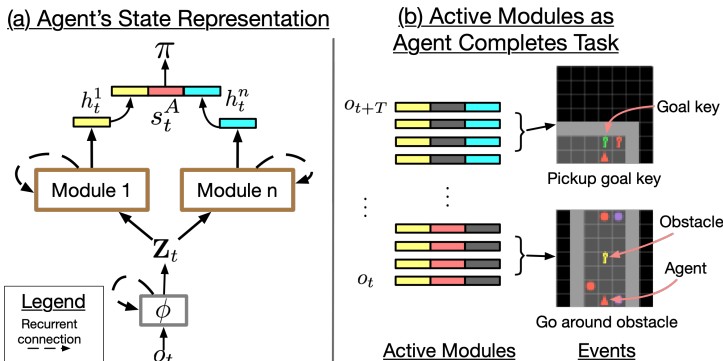

Figure 2: **Overview of FARM**. (a) FARM learns an agent state representation that is distributed across $n$ recurrent modules. (b) By distributing agent state across multiple modules, FARM is able to represent different object-centric task regularities, such as navigating around obstacles or picking up goal keys, across subsets of modules. We hypothesize that this enables a deep RL agent to flexibly recombine its experience for generalization. See §4 for details on the architecture and §5.3.1 for supporting analysis.

action and reward, $q_{t-1}^k = [h_{t-1}^k, a_{t-1}, r_{t-1}]$. The query is used to attend to observation features via a dynamic feature attention mechanism $u_t^k = f_{\texttt{att}}^k(\boldsymbol{Z}_t, q_{t-1}^k)$. The query is also used to retrieve information from other modules with a transformer-style attention mechanism $\nu_t^k = f_{\texttt{share}}^k(s_{t-1}^A, q_{t-1}^k)$. (We explain both attention mechanisms in more detail below). Each module updates with both attention outputs to produce the next module-state $h_t^k = \eta^k(u_t^k, \nu_t^k, q_{t-1}^k)$. If a task additionally has a language description $o_{\texttt{lang}}$ (as 2 of our experiments do), the module update also updates with an embedding of this description, $z_{\texttt{lang}} = f_{\texttt{lang}}(o_{\texttt{lang}})$. Agent state is then defined by the combination of these module-states $s_t^A = [h_t^1, \ldots, h_t^n]$. We illustrate this in Figure 16 and summarize the computations below:

$$\boldsymbol{Z}_t = \phi(o_t, \boldsymbol{Z}_{t-1}) \qquad \text{obs features} \qquad (1)$$

$$q_{t-1}^k = [h_{t-1}^k, a_{t-1}, r_{t-1}] \qquad \text{query} \qquad (2)$$

$$u_t^k = f_{\texttt{att}}^k(\boldsymbol{Z}_t, q_{t-1}^k) \qquad \text{obs attention} \qquad (3)$$

$$\nu_t^k = f_{\texttt{share}}^k(s_{t-1}^A, q_{t-1}^k) \qquad \text{share info} \qquad (4)$$

$$h_t^k = \eta^k(u_t^k, \nu_t^k, q_{t-1}^k) \qquad \text{module update} \qquad (5)$$

$$s_t^A = [h_t^1, \ldots, h_t^n] \qquad \text{agent state} \qquad (6)$$

where $[\cdot]$ is an operation that concatenates input vectors into a long vector. We now describe each computation in more detail.

**Structured spatiotemporal observation features.** Our first insight is that modules can attend to features describing an object's motion if an agent learns observation features that describe both spatial and temporal regularities. An agent can accomplish this by learning structured spatiotemporal features with a recurrent observation encoder $\boldsymbol{Z}_t = \phi(x_t, \boldsymbol{Z}_{t-1}) \in \mathbb{R}^{m \times d_z}$ that share $d_z$ features across $m$ spatial positions[1]. At each spatial position, these features both describe what is there visually along with temporal information about the recent dynamics of these features. We show example toy computations in Figure 3 (b).

**Dynamic feature attention.** Our second insight is that feature attention is an expressive attention function that can focus on desired information present across all spatial positions in observation features. An agent accomplishes this by having a module predict feature coefficients that it applies to uniformly across all spatial positions in $\boldsymbol{Z}_t$ (Perez et al., 2018; Chaplot et al., 2018). We show example toy computations in Figure 3 (c). We found it useful to linearly project the features before and after using shared parameters as in Andreas et al. (2016); Hu et al. (2018). The operations are summarized below:

$$f_{\texttt{att}}^k(\boldsymbol{Z}_t, q_{t-1}^k) = \left(\boldsymbol{Z}_t W_1 \odot \sigma(W_k^{\texttt{att}} q_{t-1}^k)\right) W_2 \qquad (7)$$

---

[1]One can convert height by width observation features as follows: $\mathbb{R}^{h \times w \times d_z} \to \mathbb{R}^{hw \times d_z}$

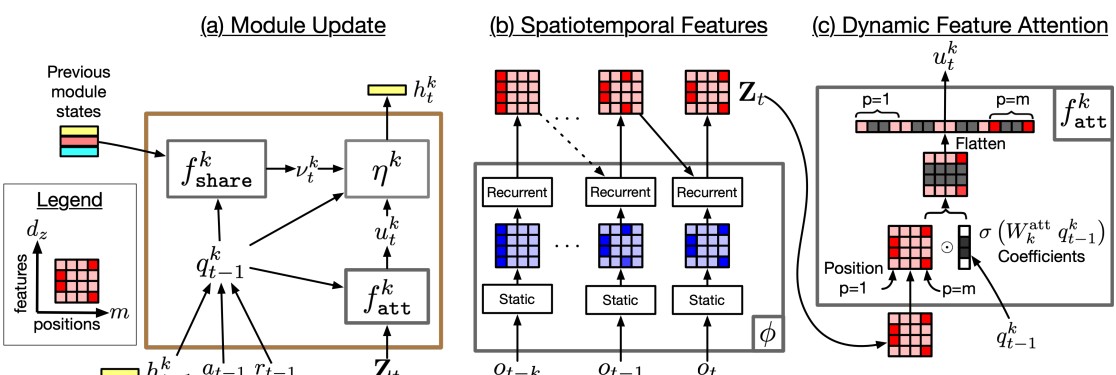

Figure 3: **Computations of FARM**. (a) Schematic of updates. See 2nd paragraph in §4 for details. (b) In order to update with features that describe both visual and temporal regularities, the agent learns structured spatiotemporal features $\boldsymbol{Z}_t \in \mathbb{R}^{m \times d_z}$ that share $d_z$ spatio-temporal features across $m$ spatial positions. Here we show toy computations where static observations features (blue) on the top and bottom row move to spatial positions to the right. The resultant spatio-temporal features (red) also include temporal information about the features (here, that the features came from leftward spatial positions). (c) $f_{\mathtt{att}}^k$ computes coefficients for features and applies them uniformly across all spatial positions. This allows the agent to attend to all spatial position that possess desired features.

where $\odot$ denotes an element-wise product over the feature dimension and $\sigma$ is a sigmoid non-linearity. Since our features capture dynamics information, this allows a module to attend to object motion (§5.1). When updating, we flatten the attention output. Flattening leads all spatial positions to be treated uniquely and allows a module to represent aspects of the observation that span multiple positions, such as 3D objects (§5.2) and spatial arrangements of objects (§5.3). Since the feature-coefficients for the next time-step are produced with observation features from the current time-step, modules can *dynamically shift* their attention when task-relevant events occur (see Figure 7, b for an example).

**Sharing information.** Similar to RIMs (Goyal et al., 2020b), before updating, each module retrieves information from other modules using transformer-style attention (Vaswani et al., 2017). We illustrate this in Figure 16 (c). We define the collection of previous module-states as $\boldsymbol{H}_{t-1} = \left[ h_{t-1}^{(1)}; \dots; h_{t-1}^{(n)}; \boldsymbol{0} \right] \in \mathbb{R}^{(n+1) \times d_h}$, where $\boldsymbol{0}$ is a null-vector used to retrieve no information. A module computes a "retrieval query" to search for information as $q_r^k = W_k^{\mathtt{que}} q_{t-1}^k \in R^{d_h}$. That module computes "retrieval keys and values" as $K^k = \boldsymbol{H}_{t-1} W_k^{\mathtt{key}} \in R^{n+1 \times d_h}$ and $V^k = \boldsymbol{H}_{t-1} W_k^{\mathtt{val}} \in R^{n+1 \times d_h}$, respectively. Each module then retrieves information as follows:

$$f_{\mathtt{share}}^k(s_{t-1}^A, q_{t-1}^k) = \mathrm{softmax}\left( \frac{q_r^k {K^k}^\top}{\sqrt{d_h}} \right) V^k. \tag{8}$$

Intuitively, the dot-product inside the softmax is computing $n+1$ scores (one for each "key"), which then form probabilities. The outer dot-product multiplies each "value" by its probability and sums them to perform soft-selection.

## 5 Experiments

In this section, we study the following questions:

1. Can FARM generalize memory to longer spatiotemporal combinations of object motions?
2. Can FARM generalize navigation towards and avoidance of 3D objects to larger environments?
3. Can FARM generalize memory of goal-information to larger maps with more distractor-based obstacles?

**Baselines.** Our first baseline is a common choice for learning state-representations, a **Long Short-term Memory (LSTM)** (Hochreiter & Schmidhuber, 1997). We study two other baselines that also attend to observation features: **Attention Augmented Agent (AAA)** (Mott et al., 2019) and **Recurrent Independent Mechanisms (RIMs)** (Goyal et al., 2020b). Both employ transformer-style attention (Locatello et al., 2020; Vaswani et al., 2017) to attend to individual *spatial positions* by reducing observation features to a weighted average over spatial positions. We instead attend to *features shared across all spatial positions*. RIMs, like FARM, represents state with a set of recurrent modules. We expand on the differences between baselines in §C.1.

Table 1: Baselines.

| Method | Observation Attention | Modular State |
|--------|:---------------------:|:-------------:|
| LSTM | ✗ | ✗ |
| AAA | Spatial | ✗ |
| RIMs | Spatial | ✓ |
| FARM (Ours) | Feature | ✓ |

**Implementation details.** We implement our recurrent observation encoder, $\phi$, as a ResNet (He et al., 2016) followed by a Convolutional LSTM (ConvLSTM) (Shi et al., 2015). We implement the update function of each module with an LSTM. We used multihead-attention (Vaswani et al., 2017) for $f^k_{\text{share}}$. We trained the architecture with the IMPALA algorithm (Espeholt et al., 2018) and an Adam optimizer (Kingma & Ba, 2015). We tune hyperparameters for all architectures with the "Place X next to Y" task from the BabyAI environment (Chevalier-Boisvert et al., 2019) (§ B.2). We expand on implementation details in §D. For details on hyperparameters, see §E.

## 5.1 Generalizing memory to more object motions

We study this with the "Ballet" grid-world (Lampinen et al., 2021) shown in Figure 1 (a). **Tasks**. The agent controls a white square that begins in the middle of the grid. There are $m$ other "ballet-dancer" objects that move with a one of 15 distinct object motions. The dances move in sequence for 16 time-steps with a 48-time-step delay in between. After all dancers finish, the agent is given a language instruction indicating the correct ballet dancer to navigate towards. All shapes and colors are randomized making motion the only feature indicating the goal object. **Observations**. The agent observes a top-down RBG image of the environment. **Actions**. The agent can move left, right, up, and down. **Reward** is 1 if it touches the correct dancer and 0 otherwise. **Tasks split**. Training tasks always consists of seeing $m = \{2, 4\}$ dancers; testing tasks always consists of seeing $m = \{8\}$ dancers. All agents learn with a sample budget of 2 billion frames. A poorly performing agent will obtain chance performance, $1/m$.

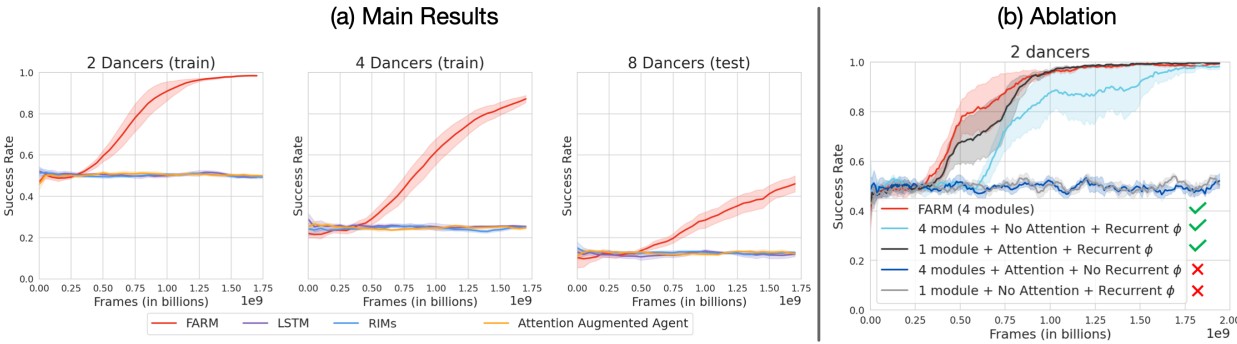

Figure 4: **FARM enables generalizing memory to longer spatiotemporal combinations of object motions**. We present the success rate means and standard errors computed using 5 seeds. (a) Only FARM is able to go above chance performance for each setting. (b) Given spatiotemporal features, we find that *either* using multiple modules *or* feature attention enables learning memory of object motions. These results suggest that spatial attention removes the benefits of using multiple modules for learning to remember object motions. Encouragingly, feature attention by itself can support it.

We present the training and generalization success rates in Figure 4. We learned spatiotemporal observation features with RIMs and AAA for a fair comparison. We found that only FARM is able to obtain above chance performance for training and testing. In order to understand the source of our performance, we ablate

using a recurrent observation encoder, using multiple modules, and using feature-attention. We confirm that a recurrent encoder is required. Interestingly, we find that either using multiple modules or using our feature-attention enables task-learning, with our feature-attention mechanism being slightly more stable.

## 5.2 Generalizing navigation with more 3D objects

Here, we study the 3D Unity environment from Hill et al. (2020) shown in Figure 1 (b). **Tasks**. The agent is an embodied avatar in a room filled with task objects and distractor objects. The agent receives a language instruction of the form "*X on Y*" —e.g., "toothbrush on bed". We partition objects into two sets as follows: pickup-able objects $O_1 = A \cup B$ and objects to place them on $O_2 = C \cup D$. **Observations.** The agent receives first-person egocentric RGB images. **Actions.** The agent has 46 actions that allow it to navigate, pickup and place objects. **Reward** is 1 if it completes the task and 0 otherwise. **Tasks split**. During training the agent sees $A \times D$ and $B \times C$ in a $4m \times 4m$ room with 4 distractors, along with $A \times C$ and $B \times D$ in a $3m \times 3m$ room with 0 distractors. We test the agent on $A \times C$ and $B \times D$ in a $4m \times 4m$ room with 4 distractors. We also train with "Go to X" and "Lift X".

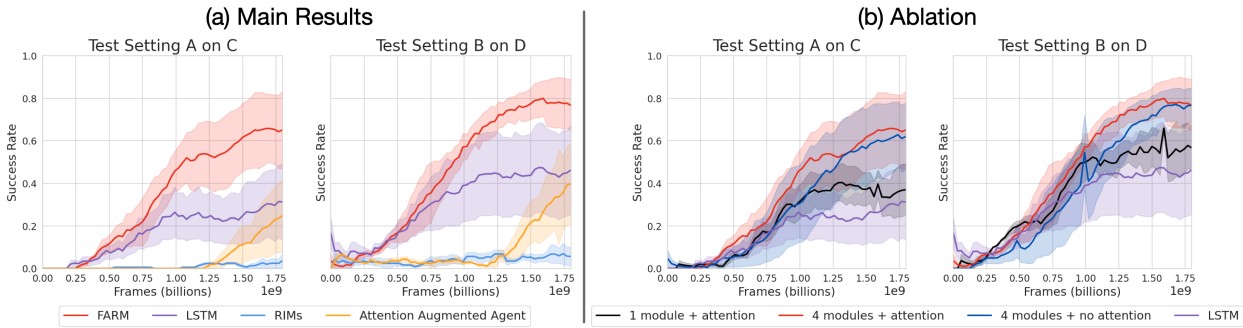

Figure 5: **FARM enables generalizing navigation towards and avoidance of 3D objects to a larger environment.** We present the success rate means and standard errors computed using 3 seeds. (a) FARM generalizes best. (b) Our performance benefits mainly come from learning multiple modules, though feature attention slightly improves performance and lowers variance. These results suggest that spatial attention interferes with reinforcement learning of 3D objects.

We present the generalization success rate in Figure 5. We find that baselines which used spatial attention learn more slowly than an LSTM or FARM. Additionally, both models that use spatial attention have poor performance until the end of training where AAA begins to improve. FARM achieves relatively good performance, achieving a success rate of 60% and 80% on the two test settings, respectively.

## 5.3 Generalizing to larger maps with more objects

To study this, we create the "keyBox" environment depicted in Figure 1 (c). **Tasks** are defined with $n$ levels. Each level is a hallway with a single box and a *key of the same color* that the agent must retrieve. The agent and the box always starts in the left-most end and the goal key always starts in the right-most end. The agent always begins in the first level. It is teleported to the next level after placing the goal key next to the goal box. The hallway for level $n$ consists of a length-$n$ sequence of $w \times w$ environment subsections. Each subsection contains $d$ distractor objects. **Observations**. The agent observes egocentric top-down images over a short segments of the hallway. **Actions**. The agent can move forward, turn left, turn right, pick up objects and drop them. **Rewards**. When completing a level, the agent gets a reward of $n/n_{\texttt{max}}$ where $n_{\texttt{max}}$ is the maximum level. **Tasks split**. Learning tasks include levels 1 to $n_{\texttt{max}} = 10$. Test tasks only use levels $2n_{\texttt{max}}$ and $3n_{\texttt{max}}$. We study two generalization settings: a *densely populated setting* with subsections of area $w^2 = 9$ and $d = 2$ distractors, and a *sparsely populated setting* with subsections of area $w^2 = 25$ and $d = 4$ distractors.

We present the generalization success rates in Figure 6. In the dense setting, we see an LSTM quickly overfits in both settings. All architectures with attention continue to improve in generalization performance as they continue training. In the dense setting, we find that FARM generalizes better (by about 20% for AAA

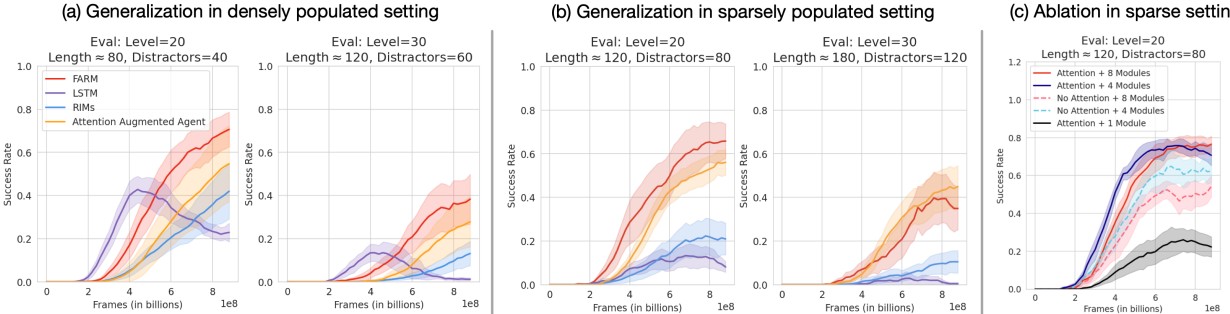

Figure 6: **FARM enables generalizing memory of goal-information and avoidance of obstacles to larger maps with more objects**. We show the the success rate mean and standard error computed using 10 seeds. (a) In the densely populated setting, FARM better generalizes to longer hallways with more distractors. (b) In the sparsely populated setting, FARM has slightly better performance than AAA for $2n_{\max}$ but comparable performance for $3n_{\max}$. (c) Using multiple modules and feature attention both improve generalization. These results suggest that spatial attention interferes with generalization benefits of learning multiple modules. Learning feature attention and multiple modules, instead, act synergistically.

and about 30% for RIMs). In the sparse setting, both RIMs and an LSTM fail to generalize above 30%. FARM generalizes better than AAA for level 20 but gets comparable performance for level 30. In some ways, this is our most surprising result since it is not obvious that either learning multiple modules or using feature attention should help with this task. In the next section we study possible sources of our generalization performance.

### 5.3.1 Analysis of state representations

We study the state representations FARM learns for categories of regularly occurring events. We collect 2000 generalization episodes in level 20. We segment these episodes into 6 categories: pickup ball, drop ball, pickup wrong key, drop wrong key, pickup correct key, and drop correct key. We study the time-series of the L2 norm of each module-state and their attention coefficients. For reference, we also show the L2 norm for the entire episode. We note that we observed consistent activity that was not captured by our simple programmatic classification of states; for example, salient activity from module 0 when the agent moved around obstacles. We show an example in Figure 7 a.

Due to space constraints, we present a subset of results in Figure 7. For all results, please see the §A. While some modules are selective for different recurring events such as attending to goal information (Figure 7, b), it seems that subsets of modules jointly represent different aspects of state. We hypothesize that this enables FARM to leverage overlapping sets of modules to store goal-information or to navigate around obstacles in a decoupled way that supports recombination. This is further supported by our ablation where we find that having 4 or 8 modules significantly outperforms using a single large module (all had about 8M params) (Figure 6, (c)). Feature attention consistently improves performance.

## 6 Discussion and conclusion

We have presented FARM, a novel state representation learning architecture for environments that have object-induced structural regularities. Our results show that we can improves an RL agent's ability to generalize to out-of-distribution tasks by having multiple modules attend to spatiotemporal features with feature attention. Specifically this enables generalizing (a) memory to longer combinations of object-motions (§5.1); (b) navigation around 3D objects to larger environments (§5.2); and (c) memory of goal information to longer sequences of obstacles (§5.3). Our ablations suggest that feature attention mainly helps with long-horion memory. Interestingly, learning multiple modules helped across all conditions (memory, obstacle-avoidance, and language learning). Our analysis suggests that learning multiple modules enables subsets to represent

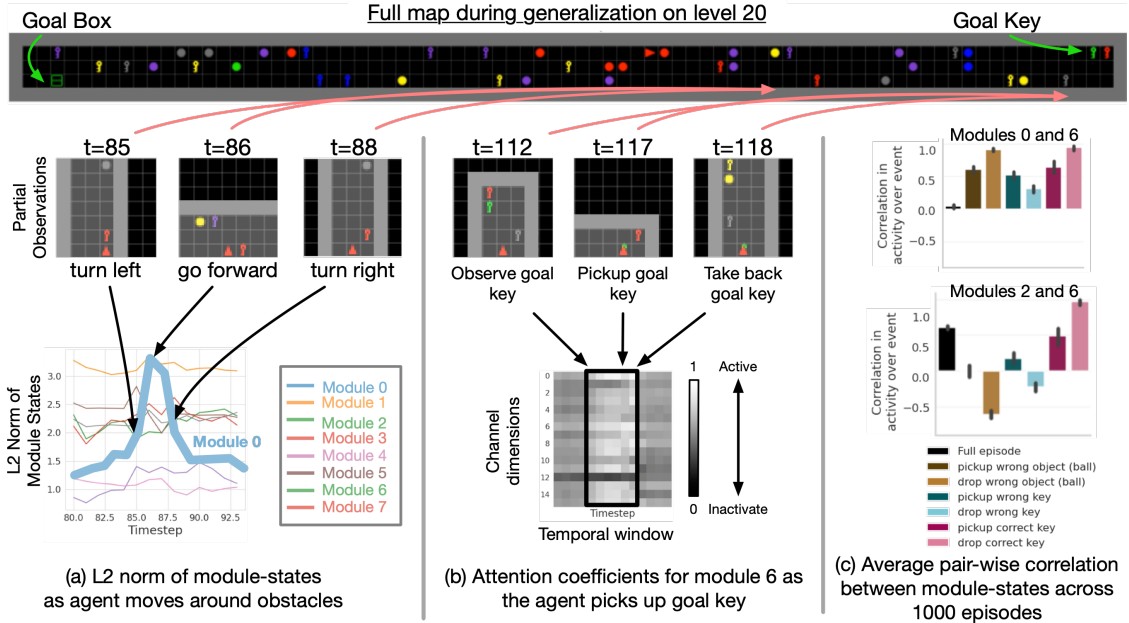

Figure 7: **We show evidence that different subsets of modules jointly represent object-induced task regularities.** (a) Module 0 commonly exhibits salient activity when the agent moves around an obstacle. (b) Module 6 activates its attention coefficients as the agent picks up the goal key. (c) Modules 2 and 6 correlate for picking up the correct key but anti-correlate for dropping the wrong object. This is similar to when neurons in word embeddings correlate for some words (e.g. man and king), but anti-correlate for other words (e.g. man and woman). In general, we find rich patterns of correlation and anti-correlation between the modules. These results suggest that FARM is representing task regularities across the modules in complicated and interesting ways. Videos of the state-activity and attention coefficients: https://bit.ly/3qCxatr.

object-centric task-relevant events in flexible ways. We hypothesize that this enables a deep RL agent to flexibly recombine its experience for generalization.

We compared FARM to other architectures that used spatial attention as a weak inductive bias for enabling objects to emerge in a state representation. We found that spatial attention hindered learning tasks with object motions and 3D objects. In the KeyBox task, spatial attention seemed to help AAA most in the sparse setting with many objects. This makes sense since spatial attention has been shown to help with distractors and the agent mainly needed to ignore objects and move forward. Interestingly, pairing spatial attention with multiple modules (RIMs) removed the benefits of both.

One limitation of FARM is that feature attention is not spatially invariant since it treat all positions as unique. Future work can look to adapt this attention for something that still describes multiple positions but in a spatially invariant way. Another limitation of FARM is the length of temporal regularities it can capture. Transformers (Vaswani et al., 2017) have shown strong performance for representing long sequences. An interesting next-step might be to integrate FARM with a transformer. We hope that our work contributes to future RL algorithms that leverage weak inductive biases for capturing object-centric task regularities.

## Acknowledgments

The authors would like to thank David Abel, Rosemary Ke, Christos Kaplanis, Tony Creswell, Agnelos Filos, Satinder Singh, Honglak Kee, and Richard Lewis for feedback on these ideas. The authors would also like to thank anonymous reviewers for their helpful feedback in making this work more accessible to a broader audience.

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
