# OpenReview forum: "Feature-Attending Recurrent Modules for Generalization in Reinforcement Learning"
_TMLR — Accepted by TMLR_

### Review · Reviewer_5aam · 2023-02-21

**Summary Of Contributions:**

This paper presents Feature-Attending Recurrent Modules (FARM) that learns state representations by introducing multiple recurrent modules that update their internal states with feature-wise attention. The main difference to prior work is that prior work uses spatial attention to learn object-centric representations but FARM utilizes feature-wise attention. Experiments are conducted to compare with prior approaches in three different experimental setups, where ablation studies on (i) introducing multiple moduels and (ii) feature-wise attention.

**Audience:**

Yes

**Claims And Evidence:**

No

**Requested Changes:**

I have no specific concerns on the method itself and experimental results. But the paper is not self-contained so it was very difficult to parse the method and I had to read the other papers to understand what the paper intends to do and how the proposed method works exactly. I'm still confused if my understanding of the method and results is correct. For instance, it's still not clear to me why feature-wise attention should perform better than spatial attention. I think the paper needs to be improved a lot in terms of providing intuition and illustrating the method.
- It would be nice to have (i) clear illustration of the difference between spatial attention and feature-wise attention and (ii) explanation and motivating example of how feature-wise attention can be suitable for the considered setup.
- Formal description of previous approaches (AAA, RIM) written in a consistent notation to the proposed method could be helpful to understand the method.
- It's not clear what does 'attending to features' mean in Intro and Related Work before reading the method section.
- What does 'Structured' mean in Structued spatiotemporal observation features?
- In (6), what does [ ] mean? How are the states combined?

**Strengths And Weaknesses:**

Strengths
- Overall stronger performance compared to baselines
- Ablation study on analysis on the proposed components

Weaknesses
- No clear intuition of why feature-wise attention can be better than spatial attention
- Writing has room for improvement, currently it's not self-contained.

---

> ### Author Response · Authors · 2023-06-07
> **Reviewer response**
>
> Thank you for helping us increase the clarity of our work. Below we respond to specific points.
> ## It would be nice to have (i) clear illustration of the difference between spatial attention and feature-wise attention and (ii) explanation and motivating example of how feature-wise attention can be suitable for the considered setup.
> We have provided an illustration of spatial vs. feature attention in Appendix ,  Figure 17. In appendix C, we describe the differences in the computations between the two. In Figure 17, we illustrate some of the core differences and elaborate on why feature attention can be better than spatial attention.
> ## Formal description of previous approaches (AAA, RIM) written in a consistent notation to the proposed method could be helpful to understand the method.
> We have provided a formal description of the spatial attention used by AAA and RIMs in Appendix C. All attention-based baselines use equivalent architectures and differ only in their observation attention function. Figure 16 shows a unified architecture shared by all 3 methods.
> ## It's not clear what does 'attending to features' mean in Intro and Related Work before reading the method section.
> We have changed the text to make clear that this means that modules produce *masks* over the channels of observation features
> ## What does 'Structured' mean in spatiotemporal observation features?
> In many Deep RL experiments, observation features are commonly vectors where spatial information is not explicitly represented in the structure of the vector. In our setting, we represent observation features with an H*W x C matrix where different rows correspond to different positions in the observation (e.g. to different positions in the output of a CNN tensor). When we say "structured", we are highlighting that the (in our cases) spatial structure of of the observation (i.e. that it has height and width) is present in the structure of the observation features. Both spatial and feature attention can exploit this structure. With feature attention, one applies a mask across all spatial positions. With spatial attention, one selects a spatial position. We have added an illustration of this in Figure 17 which we hope makes this clear.
> ## In (6), what does [ ] mean? How are the states combined?
> We have changed the text to indicate that it means that the inner vectors are concatenated along the feature dimension.

---

### Review · Reviewer_bCvr · 2023-03-15

**Summary Of Contributions:**

This paper presents a method for generalization in object-centric RL settings. The method is based on several particular inductive biases about objects, which amount to a particular learnable featurization approach, called Feature Attending Recurrent Modules (FARM). These modules process information from new observations with attention to the prior state representation using queries derived from module-specific hidden states, and the prior action and reward. Information is then shared across across the attention outputs with another attention layer that uses the same attention keys as before. Generalization experiments are presented that compare the proposed approach to its ablated versions and several related prior works. The environments used in the experiments are presented in increasing complexity of test (generalization) conditions.

**Audience:**

Yes

**Claims And Evidence:**

Yes

**Requested Changes:**

- Please include the supplementary details.
- Please run the experiments in Sec 5.3 for 1e9 frames and recompute the figures. Ideally, run them for long enough such that we observe the point at which each method begins to overfit, or that the x-axis (number of frames seen) extends quite far without overfitting (where “quite far” here would be 1e10 frames, if feasible).
- Please make the x-axis labels consistent. In Fig 4, 5 they say “Frames (billions) … 1e9” and in Fig 6 they say “Frames (billions) … 1e8”.  I assume the latter “(billions)” is incorrect.
- Please clarify how the language input is used by the model, and how the inputs are set during training.
- Please add a description of how the baselines were implemented. Include a description of the network architectures and hyperparameters and how they related to the proposed method’s architectures and hyperparmeters. Ideally, the baselines use equivalent architectures to the proposed method’s where applicable, with hyperparameters tuned on a per-baseline basis.
- Please clarify how the success rate is computed. Is it a threshold on accumulated reward?
- Please fix typo in first sentence of abstract: “in terms of object” -> “in terms of objects”.
- Consider rephrasing “objects manifest in a multitude of ways” to plainer language.
- Change the Fig. 1 caption’s first sentence to summarize the Figure content. It currently summarizes the environments, not the Figure.

**Strengths And Weaknesses:**

## Strengths
- The claims are likely interesting to some TMLR readers.
- The claims are mostly supported by evidence.
- Enough detailed is included to make the work somewhat reproducible.
- The writing is clear.

## Weaknesses
- Some of the claims are missing some evidence: We cannot draw strong conclusions about generalization without understanding the point at which the method _fails_ to generalize. For example, Figure 6 presented overfitting results for the LSTM, which raises the question: how long does it take the reset of the proposed methods to overfit? This is needed to more fully answer the questions about generalization performance.
- The supplementary material seems to be missing. There are some references to appendices that do not work (e.g. the reference to hyperparameters in S5.2, and references in S5.3.1).
- Some ambiguities in the method and baselines are present. See the requested changes.

---

> ### Author Response · Authors · 2023-06-07
> **Reviewer response**
>
> Thank you for your suggestions. Below we describe our responses.
> ## Description of how baselines were implemented
> We have included a unified description of all methods in Appendix C, Implementation details in Appendix D, and detailed our hyperparameters in Appendix E.
> All baselines use equivalent architectures and differ only in their observation attention function.
> We tuned non-attention relevant hyper-parameters using a baseline LSTM agent that doesn't use attention at all. We tuned attention-relevant hyper-parameters on a per-baseline basis.
> ## How the language input is used by the model, and how the inputs are set during training.
> First, tokens are embedded into word embeddings and then they are fed into a GRU. The last token GRU embedding is used as the language description
> During both training and evaluation, the language embedding for the current task is given to the module update function at every time-step. We detail this in Appendix C. We also note that the KeyBox environment has no language input and the task is fully specified by the agent's observations at the left-most side of the hallway.
> ## Please run the experiments in Sec 5.3 for 1e9 and make x-labels consistent.
> We mislabeled the x-axis in the original figure. This was run for 1e9 frames. We have changed the x-label to say 100s of millions. Unfortunately, running this experiment for 1e10 frames is not feasible. 1e9 frames can take 3 days. 1e10 frames would take too long.
> ## Please clarify how the success rate is computed.
> For the Ballet and Playrooom experiments: reward is 1 upon task completion. So success rate is the average reward/episode.
> For the Keybox experiments, during generalization we only evaluate an agent in a specific level. We evaluate the agent in parallel as it is training on levels 1-10. During evaluation, the agent is tested on levels 20 and 30. In this setting, reward is 1 upon task completion so the success rate is the average reward/episode.
> ## text changes
> We fixed the typo. Thank you for pointing it out.
> ### rephrasing “objects manifest in a multitude of ways”
> we have rephrased it to "objects have many dimensions that can vary"
> ### Change the Fig. 1 caption’s first sentence to summarize the Figure content
> We change the caption to "Three environments with different structural regularities induced by objects".

---

### Review · Reviewer_G2vf · 2023-05-25

**Summary Of Contributions:**

This paper presents a new network architecture that learns state representations for downstream long-horizon tasks. The representations learned are aggregated from multiple modules and each module utilizes attention mechanism to attend to spatial-temporal features. They evaluate this method on three long-horizon tasks that require the agent to reason about object movements and navigation.

**Audience:**

Yes

**Claims And Evidence:**

Yes

**Requested Changes:**

* Including more details of the experiment setup would help readers understand if the comparison to baseline methods is fair.
* Comparison to more recent baselines such as [1] would help strengthen the experiment section.
* Ablation experiments about the dynamic feature attention module would help understand if that design decision is important.


**Strengths And Weaknesses:**

### Strengths:
* The network architecture is clearly illustrated with graphs and is easy to follow.
* The proposed method achieves better results on 3 presented tasks than the chosen baselines.
* The tasks chosen are complex and each task tests different aspects of the method.

### Weaknesses:
* Some details of the experiments are not clear
    * Are the network capacities comparable between the proposed method and baselines?
    * How are the language instructions input to the network?
*  In section 5, they mention that the baselines apply a weighted sum of observation features across spatial positions while this method attends to a selected set of features shared across all spatial positions. Is this the critical design choice that makes this method better?
*  [1] also uses a set of modules to represent the state and outperforms RIMs on a set of tasks. How does this method perform compare to [1]?


[1] Madan, Kanika, et al. "Fast and slow learning of recurrent independent mechanisms." arXiv preprint arXiv:2105.08710 (2021).

---

> ### Author Response · Authors · 2023-06-07
> **Reviewer response**
>
> Thank you for your constructive feedback.  Below we detail our response.
>
> ## **Details of the experiment setup**
> We have included a unified description of all methods in Appendix C, Implementation details in Appendix D, and detailed our hyperparameters in Appendix E.
>
> ### Are the network capacities comparable between the proposed method and baselines?
> We present the number of parameters used by each method in Appendix Table 3. Different domains lead to different hyper-parameters (sometimes simply due to the size of the observation inputs). Within a domain-specific set of results, each method has about the same capacity.
>
> ### How are the language instructions input to the network?
> First, tokens are embedded into word embeddings and then they are fed into a GRU. The last token GRU embedding is used as the language description
>
> ## **Comparison to [1]**
> We argue that the results of [1] and of this paper are orthogonal. [1] used the same kind of attention of RIMs (spatial attention) and showed that meta-learning the attention parameters improves sample-efficiency. Their findings could also translate to our method, i.e. meta-learning attention parameters may improve sample-efficiency when use feature-wise attention. While [1] studied generalization, they only studied generalization to longer tasks but they did not test generalization of memory to longer tasks or generalization of state to more distractors which our experiments study. As more of their findings were around sample-efficiency, we are hopeful that future work can combine their findings and our findings to enable agents with both improved sample-efficiency and improve generalization to tasks with long-horizon memory and robustness to distractors.
>
> ## **Ablation experiments about the dynamic feature attention module**
> We point to relevant ablations of feature attention in Figures 4 and 6. In the Ballet environment (Figure 4), we see that feature attention by itself can enable learning and generalization.
> In the KeyBox environment (Figure 6), we see that feature attention and leveraging state modules jointly enable the strongest generalization.
> We also not that cross all 3 environments, pairing a modular state function with spatial attention degrades generalization performance. In Appendix C, we expand on the computational differences between spatial attention and feature attention.

---

### Author Response · Authors · 2023-06-07
**Joint response to all authors**

We thank the reviewers for their very constructive feedback which we have used to improve the clarity of our paper.  We are encouraged that reviewers found we had good performance (G2vf, 5aam) on tasks that test different aspects of the method (G2vf), clear writing (bCvr) and network illustration (G2vf), and make claims that maybe of interest to TMLR researchers (bCvr).

The biggest change we made was to add and update our supplementary material in response to reviewer questions. Most relevant to these reviews is Appendix C, where we have put a unified descriptions of baseline methods. We provide a diagram of the network architecture shared by all methods in Figure 16. We have also provided a clear toy illustration of how updating with spatial attention differs from updating with feature attention in Figure 17.

---

### Decision · Action_Editors · 2023-08-07

**Recommendation:** Accept with minor revision

**Comment:**

The first round of reviews raised concerns largely with lack of clarity in technical exposition, missing supplementary material, and asking for clarifications on relationship to some prior work, and the limits of generalization with the proposed approach.


All reviewers were satisfied with the author responses. After examination of the reviews and responses, and a less comprehensive reading through the paper, I find the proposed approach and findings interesting and valuable, and all key reviewer concerns satisfactorily addressed.

I recommend accepting the paper after incorporating the changes made during the revision (highlighted in red).


**Audience:**

This paper is likely to be of interest to all researchers in reinforcement learning techniques, particularly for physically grounded applications such as robotics, where objects are most relevant. The relevant audience is thus a substantial fraction of TMLR's target readership.



**Claims And Evidence:**

The paper claims a novel approach for generalization in object-centric RL tasks, by exploiting various inductive biases about objects in a feature learning algorithm.

The experiments in the paper reasonably validate the key claims through ablations and comparisons to baseline approaches on three environments.